# Adverse Events Associated with Nirmatrelvir/Ritonavir: A Pharmacovigilance Analysis Based on FAERS

**DOI:** 10.3390/ph15121455

**Published:** 2022-11-24

**Authors:** Meng Li, Qing-Song Zhang, Xin-Ling Liu, Hui-Ling Wang, Wei Liu

**Affiliations:** 1School of Pharmaceutical Sciences, Zhengzhou University, Zhengzhou 450001, China; 2School of Pharmaceutical Sciences, Zhengzhou Railway Vocational and Technical College, Zhengzhou 450002, China

**Keywords:** nirmatrelvir/ritonavir, COVID-19, pharmacovigilance, FAERS, adverse drug reaction

## Abstract

Nirmatrelvir/ritonavir is approved for the treatment of adults and pediatric patients with mild to moderate COVID-19, but information on adverse events associated with its use is limited. We aim to evaluate adverse events with potential risk for nirmatrelvir/ritonavir using the FDA Adverse Event Reporting System (FAERS). Disproportionality analysis was performed using the reporting odds ratio (ROR) method, and subset analysis based on patient age and gender, as well as sensitivity analysis restricting the type of reporter to healthcare professionals. Nirmatrelvir/ritonavir was the most commonly reported COVID-19 drug, and 87.66% of the outcomes were non-serious. The most frequently reported events were disease recurrence (40.43%), dysgeusia (17.55%), and diarrhea (8.80%). In disproportionality analysis, the use of nirmatrelvir/ritonavir was significantly associated with disease recurrence (ROR: 212.01, 95% CI: 162.85–276.01), whereas no signal of disease recurrence was detected for any other COVID-19 drug. Disease recurrence (ROR: 421.38, 95% CI: 273.60–648.99) was more significant when limiting the reporter type to healthcare professionals. No significant differences in adverse event reports were found based on patient gender or age. Our study confirms that the risk of serious adverse events is low with nirmatrelvir/ritonavir, but its association with disease recurrence should not be ignored.

## 1. Introduction

The coronavirus disease 2019 (COVID-19) pandemic caused by severe acute respiratory syndrome coronavirus 2 (SARS-CoV-2) is one of the largest public health events of the 21st century, with an overall hospital mortality rate of approximately 15% to 20% [1]. Although effective COVID-19 vaccines have been developed and authorized globally at an unprecedented rate [2], the treatment options available for some infected individuals remain limited. Therefore, oral antivirals that can be used to prevent COVID-19 are urgently needed to prevent the progression of infection to more severe disease, hospitalization, and death [3].

Nirmatrelvir/ritonavir (Paxlovid) is a new oral antiviral agent manufactured by Pfizer for the treatment and post-exposure prophylaxis of COVID-19. Nirmatrelvir is a SARS-CoV-2 main protease (Mpro) inhibitor that blocks viral replication by inhibiting SARS-CoV-2 Mpro activity [4,5]. Ritonavir is an HIV protease inhibitor and is considered an effective agent for the management of HIV-1 infection [6,7]. Meanwhile, ritonavir, a CYP3A4 inhibitor, is inactive against SARS-CoV-2 Mpro, but coadministration with nirmatrelvir enhances the pharmacokinetics of nirmatrelvir [4,5]. On 22 December 2021, nirmatrelvir/ritonavir received an emergency use authorization in the USA for the treatment of mild-to-moderate COVID-19 in adults and pediatric patients (12 years of age and older weighing at least 40 kg) with positive results of direct SARS-CoV-2 viral testing, and who are at high risk for progression to severe COVID-19, including hospitalization or death [8]. Nirmatrelvir/ritonavir was subsequently authorized in the United Kingdom (31 December 2021) [9], Canada (17 January 2022) [10], Australia (20 January 2022) [11], and the European Union (28 January 2022) [12].

Currently, nirmatrelvir/ritonavir is widely used in the clinic. Some studies have confirmed that nirmatrelvir/ritonavir reduces the risk of hospitalization or death due to COVID-19 infection in high-risk adult populations [3]. However, information on adverse events associated with nirmatrelvir/ritonavir is limited, with known common adverse events including dysgeusia and diarrhea, and more clinical data are needed to identify unreported serious and unexpected adverse events [3,5,13]. COVID-19 rebound has also been reported in a subset of treated individuals with initial symptom improvement, but most studies have included only a small number of cases, and whether it is a clinical phenomenon unique to nirmatrelvir/ ritonavir is unclear [14,15]. Therefore, the use of post-marketing data to detect potential adverse drug reaction signals is crucial. The aim of our study was to describe the adverse event profile of nirmatrelvir/ritonavir using the FDA Adverse Event Reporting System (FAERS) database and to identify potential risks between nirmatrelvir/ritonavir and COVID-19 cases.

The graphical workflow of this article is shown in Figure 1.

## 2. Results

During the study period, we retrieved a total of 31,491 adverse event reports from the COVID-19 emergency use authorization (EUA) FAERS public dashboard, for a total of 30,001 reports after the removal of duplicate reports. Of these, 113 reports were excluded due to the suspected inclusion of two or more COVID-19 drugs (except combination therapies) in the treatment. Finally, we included a total of 29,888 adverse event reports: 11,997 adverse event reports with nirmatrelvir/ritonavir as the suspected drug and 17,891 adverse event reports with other COVID-19 drugs as the suspected drug. Nirmatrelvir/ritonavir was the most commonly reported COVID-19 drug (40.14%), see Table 1.

### 2.1. Characteristics of Adverse Event Reports

The characteristics of reports for nirmatrelvir/ritonavir and other drugs are shown in Table 2. The male-to-female ratio was 0.57 in the nirmatrelvir/ritonavir group and 1.02 in the other drugs group. Both groups had a higher proportion in the ≥65 years age group, with median ages of 60 and 62, respectively. Reports in the nirmatrelvir/ritonavir group were mainly from consumers (71.20%) while reports in the other drugs’ groups were mainly from healthcare professionals (82.44%). The reporting countries for both groups were concentrated in the United States. In the assessment of outcomes, the majority of reported outcomes in the nirmatrelvir/ritonavir group were non-serious (87.66%), and the other drugs group reported more death (10.23% vs. 0.49%), life threatening (4.37% vs. 0.51%), required intervention (4.74% vs. 0.24%), and hospitalizations (24.85% vs. 2.83%) than the nirmatrelvir/ritonavir group.

### 2.2. Adverse Drug Events

The most common adverse events associated with the use of nirmatrelvir/ritonavir are shown in Table 3. Among them, disease recurrence (40.43%) was the most frequent adverse event except for COVID-19. The clinical consequences of patients who experienced disease recurrence were counted (Table 4), with relatively few serious consequences, such as death (0.02%) and hospitalization (0.93%), and most consequences were non-serious (96.97%). Dysgeusia (17.55%), diarrhea (8.80%), nausea (5.31%), headache (4.77%), pyrexia (2.99%), vomiting (2.88%) and malaise (2.76%) were reflected in the drug labels. According to the System Organ Classification to which the preferred term (PT) belonged, most of the adverse reactions belonged to general disorders and administration site conditions, nervous system disorders, gastrointestinal disorders, and respiratory, thoracic and mediastinal disorders. According to the Common Terminology Criteria for Adverse Events (CTCAE), most adverse events are of the highest grade 2 or 3, except for those not covered by the CTCAE.

### 2.3. Disproportionality Analysis

Figure 2 shows the signals of adverse events associated with nirmatrelvir/ritonavir use. The results showed that the use of nirmatrelvir/ritonavir was significantly associated with disease recurrence (ROR: 212.01, 95% CI: 162.85–276.01), and similar adverse events were symptom recurrence (ROR: 62.10, 95% CI: 31.95–120.68) and therapeutic product effect incomplete (ROR: 55.50, 95% CI: 24.56–125.41), but no positive signals were detected for such adverse events for any of the other COVID-19 drugs.

Figure 3 and Figure 4 show the disproportionality results based on patient age and gender, respectively, with the top four adverse events being disease recurrence, dysgeusia, product taste abnormal, and symptom recurrence. Comparing the differences in adverse events by age, the differences for patients < 65 years were flatulence, upper respiratory tract congestion, and sleep disorder, and for patients ≥ 65 years were product prescribing error, product use complaint, and nasopharyngitis. Comparing the differences in adverse events by gender, the differences were product prescribing error, respiratory congestion and nasopharyngitis in male patients and product use complaints, upper respiratory congestion and flatulence in female patients.

Figure 5 shows the results of the sensitivity analysis restricting the type of reporter to healthcare professionals. Overall, the results of the sensitivity analysis were generally consistent with the primary analysis and nirmatrelvir/ritonavir showed a higher correlation with disease recurrence (ROR: 421.38, 95% CI: 273.60–648.99).

## 3. Discussion

Nirmatrelvir/ritonavir is approved for the treatment of adult or pediatric patients at high risk of progression to severe COVID-19, but clinical experience and evidence for this new oral antiviral drug is limited. We conducted a pharmacovigilance study using COVID-19 EUA FAERS public dashboard data to establish the relationship between nirmatrelvir/ritonavir and adverse events.

The results of our study showed that disease recurrence accounted for 40.43% of nirmatrelvir/ritonavir adverse event reports. The disproportionality analysis conducted in our study confirmed that nirmatrelvir/ritonavir was significantly associated with disease recurrence, but no signal was detected for disease recurrence for any of the other COVID-19 drugs. One study concluded that neither nirmatrelvir resistance nor lack of neutralizing immunity were possible causes of the observed relapses, and that the most likely cause was inadequate drug exposure or duration due to individual pharmacokinetics [16]. Another study found that disease recurrence after nirmatrelvir/ritonavir treatment of early COVID-19 infection was associated with high viral load in individuals, and the results similarly supported that drug resistance was not an important cause of relapse [17]. According to the Centers for Disease Control and Prevention, a brief return of symptoms may be part of the natural history of SARS-CoV-2 infection in some individuals, independent of Paxlovid treatment and regardless of vaccination status [15]. Based on the results of this study, the clinical outcome of most patients with disease recurrence was non-serious, but close monitoring of patients is still recommended.

Dysgeusia, diarrhea, nausea, headache, fever, vomiting and malaise are reflected in the nirmatrelvir/ritonavir label [18,19]. Among them, dysgeusia and diarrhea occurred at a higher frequency. This is consistent with the results of the EPIC-HR trial [3]. According to the ritonavir label, gastrointestinal disorders (including diarrhea, nausea, and vomiting), nervous system disorders (including paresthesia and dysgeusia), and fatigue were the most frequently reported adverse drug reactions among patients receiving ritonavir alone or in combination with other antiretrovirals [20]. Notably, taste disorder is also a common symptom in patients with mild to moderate COVID-19 [21,22,23].

Respiratory, thoracic, and mediastinal disorders such as cough, nasal congestion, oropharyngeal pain, and rhinorrhea are more frequent, as well as upper respiratory tract congestion and sneezing are more evident in the disproportionality analysis. COVID-19, a viral-induced respiratory disease, has cough as its second most common symptom [24]. Nasal congestion and rhinorrhea are clinical characteristics in patients with mild COVID-19 [25,26]. Also, cough and oropharyngeal pain are reflected in the ritonavir label [20]. However, these adverse events are not reflected in the nirmatrelvir/ritonavir label and are lacking in reports in the current study.

Our study also evaluated potential differences in adverse event reporting based on age and sex and did not find significant differences in age and sex, especially when organ systems were considered as a whole. For example, product prescribing error and respiratory tract congestion in male patients and product use complaint and upper respiratory tract congestion in female patients were classified as injury, poisoning, and procedural complications, and respiratory, thoracic, and mediastinal disorders, respectively.

Unlike other COVID-19 drugs, for which the primary reporter were healthcare professionals, nirmatrelvir/ritonavir had a higher rate of consumer reporting. Considering that reports reported by healthcare professionals are more specialized than consumers, we conducted a sensitivity analysis limiting the reporters to healthcare professionals. The results showed that disease recurrence had a higher signal value in the sensitivity analysis than in the primary analysis, and also further demonstrated the strong association between disease recurrence and nirmatrelvir/ritonavir.

Overall, infected patients on nirmatrelvir/ritonavir had a lower risk of developing more serious disease, hospitalization, or death than did patients on other COVID-19 drugs, and most adverse events were grade 2–3. In the EPIC-HR trial, nirmatrelvir/ritonavir reduced hospitalization or mortality in COVID-19 ambulatory patients by 89% compared with the control group, and both groups had similar rates of adverse events during treatment, with the nirmatrelvir/ritonavir group having fewer grade 3 or 4 adverse events (4.1% vs. 8.3%) and serious adverse events (1.6% vs. 6.6%) than the placebo group [3]. A large integrated health care system-based study found that <1% of patients experienced COVID-19-related hospitalizations or emergency department events 5–15 days after receiving Paxlovid [27]. In a cohort study of early COVID-19 outpatients, the risk of hospitalization was reduced by 45% in patients receiving a nirmatrelvir/ritonavir prescription [28]. The result of another large retrospective cohort study similarly showed that nirmatrelvir/ritonavir was associated with a significant reduction in severe morbidity or mortality from COVID-19 [29].

The current study has several limitations. FAERS is a spontaneous reporting system, but not all adverse events regarding drugs are reported to the FDA, so under- or over-reporting and missing information are inevitable. Furthermore, although our study revealed an association between nirmatrelvir/ritonavir and potential risk in patients with COVID-19, a causal relationship between drug use and adverse events cannot be determined solely from FAERS data and needs to be confirmed by clinical studies.

## 4. Materials and Methods

FAERS contains adverse event reports submitted to the FDA by healthcare professionals, consumers, and the pharmaceutical industry, and data is available through quarterly data files or public dashboards. In response to the COVID-19 pandemic, the FDA launched the FAERS Public Dashboard for COVID-19 EUA products, including adverse event reports for drugs and therapeutic biological products used under the COVID-19 EUA, and updated weekly. In the COVID-19 EUA FAERS Public Dashboard, drug effects are categorized as “suspected drug” and “concomitant drug”, and adverse events use the PT of the Medical Dictionary for Regulatory Activities (MedDRA, access date: 28 October 2022) to encode.

### 4.1. Data Collection and Processing

We extracted all reports in the COVID-19 EUA FAERS public dashboard up to 29 July 2022. To avoid double counting of multiple versions of reports, we judged potential duplicate reports with different case IDs based on suspected product name, suspected product active ingredient, reactions, patient sex, age, reporting country, event date, and concomitant product name, and reports with matching information in the above eight fields were considered duplicates and went in for de-duplication. In addition, reports containing two or more COVID-19 drugs (except combination therapies) in the suspected drug were removed in order to ensure the association of adverse events with the suspected drug. Combination therapies were standardized with FDA EUA. Serious adverse events were defined as those resulting in death, life-threatening, hospitalization (initial or prolonged), disability or permanent damage, congenital anomaly, need for intervention to prevent permanent impairment or damage, and other serious adverse events.

### 4.2. Statistical Analysis

Descriptive statistical analysis included patient sex, age, reporter type, reporting country, and adverse event outcome. Categorical variables are reported as frequencies and percentages, and continuous variables are reported as medians, frequencies, and percentages.

We used the reporting odds ratio (ROR) method for disproportionality analysis, and calculations were performed based on a 2-by-2 contingency table (Table 5). Reports with nirmatrelvir/ritonavir as the suspected drug were included in the target drug group; reports without nirmatrelvir/ritonavir as the suspected drug were included in the other drug group. A risk signal was considered significant when the total number of drugs and adverse events was ≥3 and the lower limit of the 95% CI of the ROR was >1 [30].

To assess the robustness of the main analyses, we also performed subset analyses based on patient age and gender, as well as sensitivity analyses that restricted the type of reporter to healthcare professionals. All data were processed and analyzed using R (version 4.1.0, R Development Core Team).

## 5. Conclusions

This is the first pharmacovigilance study to determine the potential risk between nirmatrelvir/ritonavir and COVID-19 patients based on real FAERS data, but it needs to be confirmed by further clinical studies.

Overall, nirmatrelvir/ritonavir had a lower risk of serious adverse events, with 87.66% of outcomes being non-serious, reducing the risk of infection progressing to more serious disease, hospitalization, and death. No significant age or gender differences were found in the adverse event reports of these patients. However, nirmatrelvir/ritonavir was significantly associated with disease recurrence (ROR: 212.01, 95% CI: 162.85–276.01), which was more significant when limiting the reporter type to healthcare professionals (ROR: 421.38, 95% CI: 273.60–648.99). In contrast, no signal for disease recurrence was found for other COVID-19 drugs. Therefore, the association between nirmatrelvir/ritonavir and disease recurrence should not be overlooked.

## Figures and Tables

**Figure 1 pharmaceuticals-15-01455-f001:**
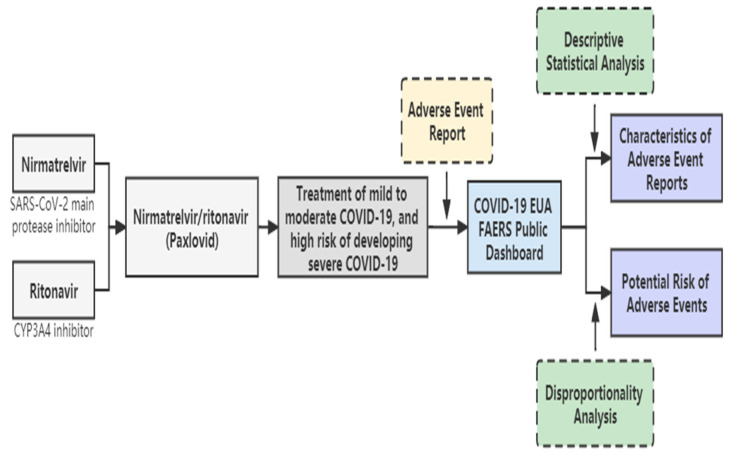
Graphical workflow of this article.

**Figure 2 pharmaceuticals-15-01455-f002:**
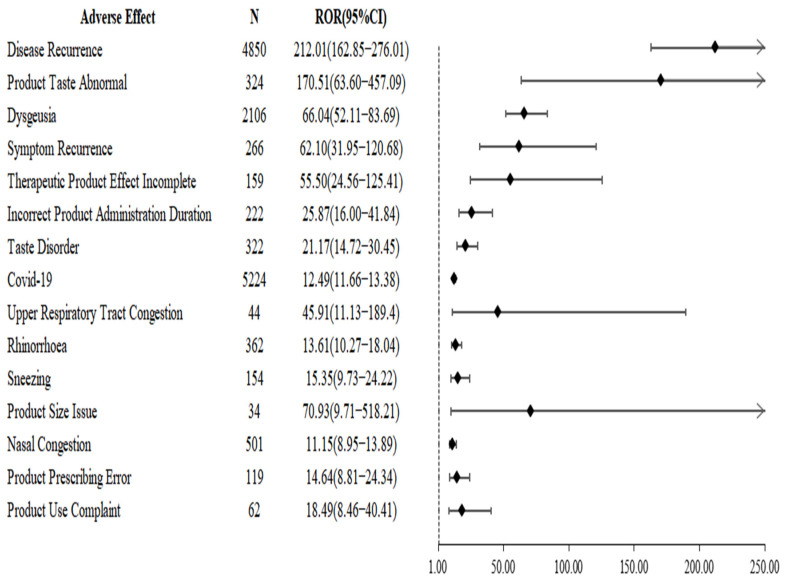
Signals of adverse events associated with nirmatrelvir/ritonavir use.

**Figure 3 pharmaceuticals-15-01455-f003:**
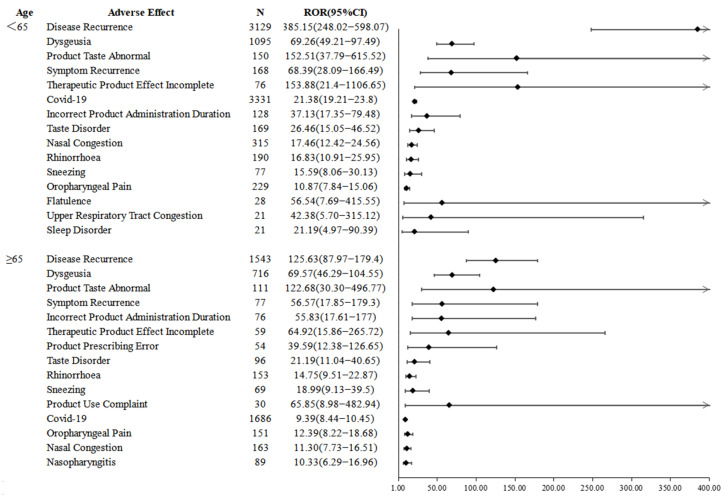
Signals of adverse events associated with nirmatrelvir/ritonavir use based on patient age.

**Figure 4 pharmaceuticals-15-01455-f004:**
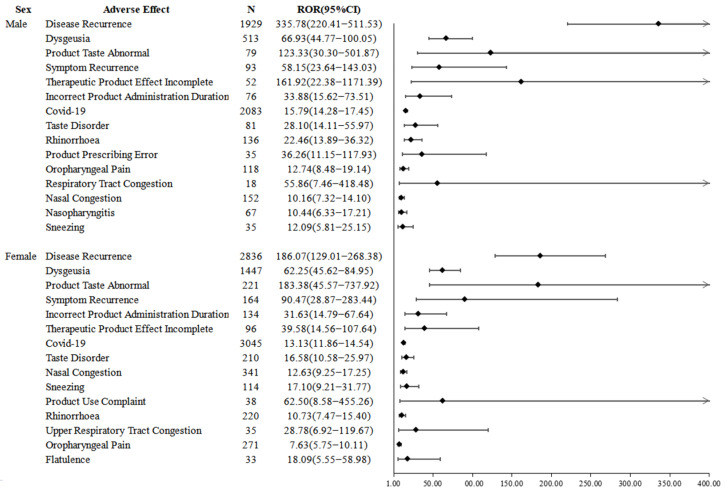
Signals of adverse events associated with nirmatrelvir/ritonavir use based on patient sex.

**Figure 5 pharmaceuticals-15-01455-f005:**
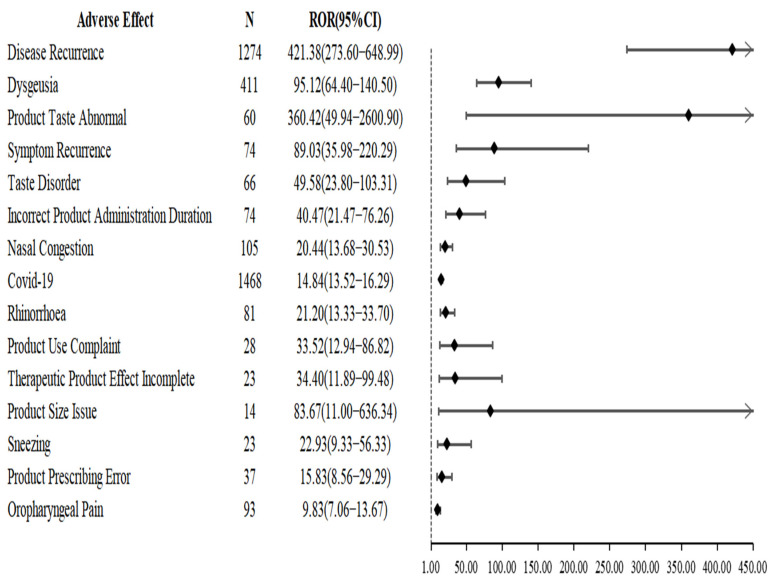
Results of sensitivity analysis associated with nirmatrelvir/ritonavir use.

**Table 1 pharmaceuticals-15-01455-t001:** Top 10 COVID-19 drugs reported as suspected drugs in FAERS.

Suspected Drug	N	%
Nirmatrelvir/Ritonavir	11,997	40.14
Casirivimab/Imdevimab	4478	14.98
Remdesivir	4047	13.54
Bamlanivimab	3889	13.01
Bamlanivimab/Etesevimab	1968	6.58
Sotrovimab	1032	3.45
Baricitinib	652	2.18
Bebtelovimab	621	2.08
Cilgavimab/Tixagevimab	466	1.56
Tocilizumab	253	0.85

**Table 2 pharmaceuticals-15-01455-t002:** Characteristics of adverse event reports using nirmatrelvir/ritonavir or other COVID-19 drugs as suspected drugs.

Characteristics	Nirmatrelvir/Ritonavir (N = 11,997)		Other COVID-19 Drugs (N = 17,891)	
	N	%	N	%
Sex				
Female	7138	59.50	8430	47.12
Male	4082	34.03	8590	48.01
Not Specified	777	6.48	871	4.87
Age				
Median	60		62	
<18	27	0.23	331	1.85
18–44	2357	19.65	3272	18.29
45–64	3887	32.40	5309	29.67
≥65	4132	34.44	6864	38.37
Not Specified	1594	13.29	2115	11.82
Reporter Type				
Consumer	8542	71.20	2551	14.26
Healthcare Professional	3415	28.47	14,749	82.44
Not Specified	40	0.33	591	3.30
Country				
US	10,553	87.96	13,245	74.03
Other Countries	818	6.82	718	4.01
Not Specified	626	5.22	3928	21.96
Outcome				
Death	59	0.49	1831	10.23
Life Threatening	61	0.51	782	4.37
Required Intervention	29	0.24	848	4.74
Disabled	48	0.40	83	0.46
Hospitalizations	339	2.83	4446	24.85
Congenital Anomaly	0	0.00	5	0.03
Other Outcomes	944	7.87	4658	26.04
Non-Serious	10,517	87.66	5238	29.28

**Table 3 pharmaceuticals-15-01455-t003:** Common adverse events reported to FAERS with nirmatrelvir/ritonavir as a suspected drug.

Preferred Terms	System Organ Classes	N	%
COVID-19	Infections and infestations	5224	43.54
Disease Recurrence	General disorders and administration site conditions	4850	40.43
Dysgeusia ^a^	Nervous system disorders	2106	17.55
Diarrhoea	Gastrointestinal disorders	1056	8.80
Nausea	Gastrointestinal disorders	637	5.31
Cough ^b^	Respiratory, thoracic and mediastinal disorders	632	5.27
Fatigue ^b^	General disorders and administration site conditions	579	4.83
Headache ^b^	Nervous system disorders	572	4.77
Nasal Congestion ^b^	Respiratory, thoracic and mediastinal disorders	501	4.18
Oropharyngeal Pain ^b^	Respiratory, thoracic and mediastinal disorders	409	3.41
Rhinorrhoea	Respiratory, thoracic and mediastinal disorders	362	3.02
Pyrexia	General disorders and administration site conditions	359	2.99
Vomiting ^c^	Gastrointestinal disorders	346	2.88
Incorrect Dose Administered	Injury, poisoning and procedural complications	345	2.88
Malaise ^b^	General disorders and administration site conditions	331	2.76

^a^ The adverse event is up to level 2 in CTCAE. ^b^ The adverse event is up to level 3 in CTCAE. ^c^ The adverse event is up to level 5 in CTCAE.

**Table 4 pharmaceuticals-15-01455-t004:** Clinical outcomes of patients with disease recurrence after nirmatrelvir/ritonavir use.

Outcomes	N	%
Death	1	0.02
Life Threatening	4	0.08
Disabled	9	0.19
Hospitalization	45	0.93
Other Outcomes	88	1.81
Non-Serious	4703	96.97

**Table 5 pharmaceuticals-15-01455-t005:** 2 × 2 Contingency table for disproportionality analysis.

	Target Adverse Event	All other Adverse Event	Total
Target drug	n11	n12	n1+
All other drugs	n21	n22	n2+
Total	n + 1	n + 2	n++

## Data Availability

FDA Adverse Event Reporting System data are available at https://www.fda.gov/drugs/questions-and-answers-fdas-adverse-event-reporting-system-faers/fda-adverse-event-reporting-system-faers-public-dashboard (accessed on 28 October 2022).

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
