# Peer review of "Adverse Events Associated with Nirmatrelvir/Ritonavir: A Pharmacovigilance Analysis Based on FAERS"

_pharmaceuticals, 2022, doi:10.3390/ph15121455_

Round 1

Reviewer 1 Report

The submitted article ‘Adverse Events Associated with Nirmatrelvir/ritonavir:
A Pharmacovigilance Analysis Based on FAERS’ is interesting and good article. This is a kind of rare studies, hence is very desire. It will be appropriate for PHARMACEUTICALS (MDPI). Below I pointed most of mistakes and matters for explanation.

1.      In introduction the emphasis should be placed on justification

2.      In introduction, there is a lack basic information about application of ritonavir in AIDS treatment – this is a big disadvantage; please cite also PMID: 20836579 and PMID: 28614778

3.      The good idea should be summary as graphical workflow in introduction

4.      Conclusion part should be more informative (more details, perhaps in brackets?)

5.      Please include advances and disadvantages of your studies in conclusions

I totally agree that this is very important subject, and it is very important for publication in MDPI. I will recommend this article for publication in PHARMACEUTICALS after minor revision.

Reviewer 2 Report

By utilizing data obtained from the FDA adverse event reporting system, the authors preformed a disproportionality analysis to evaluate the risk of adverse events resulting from nirmatrelvir/ritonavir in the treatment of patients with COVID-19. Their principal finding demonstrated a significant signal for the association of disease recurrence and treatment with nirmatrelvir/ritonavir (as opposed to all other treatments, which had no signal).

General comments:

·      The manuscript was polished, concise, and well written.

·      Results were supported by using subset(age/gender) and sensitivity (AE reporter only healthcare professionals) analysis.

o   As FAERS is a spontaneous reporting system, confirmation with an external database would have demonstrated that the results were not a product of or biased by the reporting system.

·      Data interpretational was rational
